# Upregulation of DR5 and Downregulation of Survivin by IITZ-01, Lysosomotropic Autophagy Inhibitor, Potentiates TRAIL-Mediated Apoptosis in Renal Cancer Cells via Ubiquitin-Proteasome Pathway

**DOI:** 10.3390/cancers12092363

**Published:** 2020-08-21

**Authors:** Sk Abrar Shahriyar, Seung Un Seo, Kyoung-jin Min, Peter Kubatka, Do Sik Min, Jong-Soo Chang, Dong Eun Kim, Seon Min Woo, Taeg Kyu Kwon

**Affiliations:** 1Department of Immunology, School of Medicine, Keimyung University, 1095 Dalgubeoldaero, Dalseo-Gu, Daegu 42601, Korea; sksy.kmu@gmail.com (S.A.S.); sbr2010@hanmail.net (S.U.S.); 2New Drug Development Center, Daegu-Gyeongbuk Medical Innovation Foundation, 80 Chembok-ro, Dong-gu, Daegu 41061, Korea; kjmin@dgmif.re.kr; 3Department of Medical Biology, Jessenius Faculty of Medicine, Comenius University in Bratislava, 03601 Martin, Slovak; kubatkap@gmail.com; 4Department of Experimental Carcinogenesis, Division of Oncology, Biomedical Center Martin, Jessenius Faculty of Medicine, Comenius University in Bratislava, 03601 Martin, Slovak; 5College of Pharmacy, Yonsei University, 85 Songdogwahak-ro, Yeonsu-gu, Incheon 21983, Korea; minds@yonsei.ac.kr; 6Department of Life Science and Chemistry, College of Natural Science Daejin University, Pochon-shi, Kyeinggido 11150, Korea; jchang@daejin.ac.kr; 7Department of Otolaryngology, School of Medicine, Keimyung University, 1095 Dalgubeoldaero, Dalseo-Gu, Daegu 42601, Korea; entkde@dsmc.or.kr

**Keywords:** IITZ-01, TRAIL, DR5, survivin, USP9X

## Abstract

Tumor necrosis factor-related apoptosis-inducing ligand (TRAIL) selectively is able to increase apoptosis in cancer cells as agent with minimum toxicity to noncancerous cells. However, all cancer cells are not sensitive to TRAIL-induced apoptosis. In this study, we showed the sub-lethal concentrations of a lysosomotropic autophagy inhibitor, IITZ-01, sensitizes cancer cells (renal, lung, and breast carcinoma) to TRAIL-induced apoptosis through DR5 upregulation and survivin downregulation through ubiquitin-proteasome pathway. Knockdown of DR5 or overexpression of survivin inhibited combined treatment with IITZ-01 and TRAIL-induced apoptosis. IITZ-01 downregulated protein expression of Cbl, ubiquitin E3 ligase, and decreased expression level of Cbl markedly led to increase DR5 protein expression and TRAIL sensitivity. Moreover, IITZ-01 decreased expression level of survivin protein via downregulation of deubiquitinase ubiquitin-specific protease 9X (USP9X) expression. Taken together, these results provide the first evidence that IITZ-01 enhances TRAIL-mediated apoptosis through DR5 stabilization by downregulation of Cbl and USP9X-dependent survivin ubiquitination and degradation in renal carcinoma cells.

## 1. Introduction

Lysosomes are acidic organelles within cells that degrade and reuse macromolecules through endocytosis, autophagy and phagocytosis, and are involved in regulation of cellular homeostasis [1,2]. Lysosomotropic agents can selectively diffuse, accumulate into lysosomes, and consequently induce pH alteration, decrease of enzyme activity and inhibition of calcium signaling in lysosome [3]. In addition, several lysosomotropic drugs break down phagocytosis, endocytosis and autophagy by disrupting membrane fusion between intracellular organelles [4,5,6]. Chloroquine (CQ) and hydroxychloroquine (HCQ), well-known lysosomotropic drugs, increase lysosomal membrane permeabilization by neutralizing intracellular lysosomal pH, resulting in the induction of apoptotic cell death in cancer [7,8]. Moreover, two lysosomotropic agents can potentiate anti-cancer effect to various chemotherapeutic drugs and overcome resistance to anti-cancer drugs or irradiation in many cancer cells [9,10,11,12,13]. Therefore, lysosomotropic agents are capable of increasing therapeutic efficacy of tumors, but the underlying molecular mechanism is not clear.

IITZ-01, as benzimidazole analogs of morpholino s-triazine, has been kwon to contain lysosomotrpopic properties and anti-cancer activity [14,15]. IITZ-01 accumulates autophagosome by inhibiting autophagic flux through damaged lysosome. In addition, IITZ-01 decreases mitochondria membrane potential through downregulation of Bcl-2 and IAP family proteins followed by caspase-dependent apoptosis in breast cancer cells [16].

The tumor necrosis factor-related apoptosis-inducing ligand (TRAIL) targets malignant cells by binding to the death receptors (DRs), followed by the increase of caspase-dependent apoptosis except normal cells [17,18]. Although TRAIL is considered to be a strong therapeutic agent of malignant cells, many malignant cells have been demonstrated to be resistant to TRAIL. Therefore, many researchers have reported that combined treatment with numerous sensitizing agents is able to overcome TRAIL resistance [19,20].

In our present study, we found the effect of IITZ-01 as a novel TRAIL sensitizer. IITZ-01 induced Cbl-mediated DR5 stabilization and USP9X-dependent survivin degradation, resulted in the enhancement to TRAIL sensitivity in cancer.

## 2. Results

### 2.1. IITZ-01 Increases TRAIL Sensitivity in Renal Carcinoma Cells

IITZ-01 exerts an anticancer effect in breast, colon and prostate cancer cells [15,16]. We examined whether IITZ-01 could augment TRAIL-induced apoptosis in human renal carcinoma cells. As shown in Figure 1a, individual IITZ-01 (0.5, 1 μM) and TRAIL (30, 50 ng/mL) had no effect on apoptosis in human renal carcinoma (Caki-1 and ACHN) cells, but combined treatment of IITZ-01 and TRAIL remarkably increased the sub-G1 population and cleavage of PARP. Furthermore, IITZ-10 plus TRAIL treatment increased Annexin V/7-Aminoactinomycin D (7-AAD) double positive cells (Figure 1b). We observed typical apoptotic morphologies, including blebbing, apoptotic bodies, exemplary chromatin damage in the nuclei and DNA fragmentation in Caki-1 cells treated with IITZ-01 and TRAIL (Figure 1c). We investigated whether caspases activation plays a significant role in cell death by combined treatment with IITZ-01 plus TRAIL. Combined treatment increased the caspase-3 activity (Figure 1d). Furthermore, when we used z-VAD-fmk (z-VAD), a pan-caspase inhibitor, pretreatment of z-VAD completely abolished apoptosis and cleavage of caspase-3 by IITZ-01 plus TRAIL (Figure 1e). These results suggest that IITZ-01 enhances TRAIL-induced apoptosis in human renal carcinoma cells.

### 2.2. Effect of IITZ-01 on Expression Levels of Apoptosis-Related Proteins

To elucidate the molecular mechanisms leading to apoptosis in IITZ-01 treated Caki-1 and ACHN cells, we investigated the changes of apoptosis-related proteins expression. IITZ-01 significantly diminished survivin expression and increased DR5 expression, while other apoptosis-related proteins (Mcl-1, XIAP, Bcl-2, Bcl-xL, Bim, cIAP1, cIAP2, DR4, c-FLIP and Bax) were not affected by IITZ-01 treatment (Figure 2). These results indicate that IITZ-01 induces survivin downregulation and DR5 upregulation, resulting in the increase of TRAIL-mediated apoptosis in caspase-dependent manner.

### 2.3. Upregulation of DR5 by IITZ-01 Is Associated with TRAIL-Induced Apoptosis

DR5 plays a major role in TRAIL-mediated apoptosis [21,22], and other lysosomotropic agents induce DR5 upregulation resulting in the enhancement of TRAIL sensitivity [23]. Expectably, DR5 protein level was increased by IITZ-01 after 3 h (Figure 3a). To explore the IITZ-01-mediated DR5 upregulation at the transcriptional levels, we checked DR5 mRNA level and promoter activity. However, IITZ-01 did not alter DR5 mRNA level and promoter activity (Figure 3b,c). Next, we investigated the impact of IITZ-01 on DR5 protein stability using the cycloheximide (CHX). Combined treatment with CHX and IITZ-01 more maintained DR5 protein level compared to CHX alone (Figure 3d). Next, we investigated the importance of the DR5 upregulation in IITZ-01 plus TRAIL-induced apoptosis, Caki-1 and ACHN cells were transiently transfected with DR5 siRNA. Knockdown of DR5 significantly reduced apoptosis by the combined treatment of IITZ-01 and TRAIL (Figure 3e). Therefore, these data indicate that upregulation of DR5 expression is involved in IITZ-01 plus TRAIL-induced apoptosis.

### 2.4. Downregulation of Cbl Plays a Critical Role in DR5 Upregulation and TRAIL Sensitivity by IITZ-01

Previous studies reported that E3 ligase Cbl contributes to DR5 ubiquitination and degradation [24,25]. We also reported that CQ-mediated Cbl downregulation induce decrease of DR5 protein expression level [23]. Following these, we checked the role of Cbl in DR5 expression in IITZ-01 treated cells. IITZ-01 downregulated Cbl protein level in a time-dependent manner, but not mRNA level (Figure 4a). We further examined whether IITZ-01 can regulate Cbl stabilization. As shown in Figure 4b, IITZ-01 significantly inhibited Cbl stability compared with CHX alone. To further confirm the involvement of Cbl on DR5 upregulation by IITZ-01, we transiently overexpressed Cbl. Overexpression of Cbl markedly diminished IITZ-01 induced DR5 upregulation (Figure 4c). Downregulation of Cbl by siRNA induced TRAIL-mediated apoptosis, cleavage of PARP and DR5 upregulation in Caki-1 and ACHN cells (Figure 4d). These data demonstrate that Cbl is important role in stability of DR5 and sensitization of TRAIL by IITZ-01.

### 2.5. Survivin Degradation Is Important for IITZ-01 Plus TRAIL-Induced Apoptosis

We next investigated whether IITZ-01 modulates survivin expression at the transcriptional level. Survivin mRNA expression was not altered by IITZ-01 treatment (Figure 5a). Therefore, we investigated whether IITZ-01 reduces survivin expression at the post-translational level. Combined treatment with IITZ-01 and CHX more quickly decreased survivin protein expression rather than CHX alone (Figure 5b). In addition, MG132, a proteasome inhibitor, reversed IITZ-01-induced survivin downregulation (Figure 5c). To investigate the functional role of survivin in combined treatment IITZ-01 plus TRAIL-mediated apoptosis, we used survivin-overexpressed Caki-1 and ACHN cells. Ectopic expression of survivin prevented sub-G1 population and PARP cleavage by combined treatment (Figure 5d). Therefore, these data suggested that IITZ-01 reduces survivin expression via proteasome activity at the posttranslational level, and downregulation of survivin contributes to IITZ-01-induced TRAIL sensitization.

### 2.6. USP9X-Dependent Survivin Degradation Is Important for IITZ-01 Plus TRAIL-Induced Apoptosis

Among deubiquitinases, ubiquitin specific peptidase 9X (USP9X) and STAM-binding protein-like 1 (STAMBPL1) are involved in stabilization of survivin [26,27]. Therefore, we examined protein expression levels of USP9X and STAMBPL1 by IITZ-01. As shown in Figure 6a, IITZ-01 decreased USP9X protein expression, but not STAMBPL1. Moreover, USP9X mRNA expression was not changed by IITZ-01 (Figure 6b). Therefore, we examined post-translational regulation of IITZ-01-mediated UPS9X downregulation and found CHX plus IITZ-01 more degraded USP9X expression (Figure 6c). To examine whether USP9X physically interacts with survivin, we performed an immunoprecipitation assay by using Caki-1 cells. Immunoprecipitation assay revealed that USP9X reciprocally binds to survivin, but not control IgG (Figure 6d). To verify the functional role of USP9X in degradation of survivin by IITZ-01, we used constructs with wild-type (WT) and catalytically inactive mutant (C1566S) of USP9X. USP9X WT markedly impeded survivin degradation by IITZ-01 in Caki-1/Vector cells, whereas USP9X/C1566S still revealed IITZ-01-inhibited survivin expression (Figure 6e). Next, we examined whether USP9X regulates survivin deubiquitination in cells. Transfection of cells with HA-Ub expressing vector induced survivin ubiquitination (Figure 6f). However, ubiquitination of survivin was significantly decreased in USP9X WT-overexpressed cells. In contrast, catalytically inactive mutant USP9X increased survivin ubiquitination (Figure 6f). Taken together, our results suggest that IITZ-01 can inhibit survivin ubiquitination via downregulation of USP9X, resulted in degradation of survivin protein expression.

### 2.7. IITZ-01 Plus TRAIL Treatment Effects on Apoptosis in Various Cancerous but Not in Normal Cells

We further investigate whether IITZ-01 plus TRAIL trigger apoptosis in other carcinoma cells and normal cells. As shown in Figure 7a, combined treatment with IITZ-01 and TRAIL augmented sub-G1 population and PARP cleavage in other renal cancer cells (A498), human lung cancer cells (A549) and breast cancer cells (MCF7). Moreover, IITZ-01 upregulated DR5 and downregulated Cbl, USP9X and survivin in all examined cancer cells (Figure 7b). However, IITZ-01 plus TRAIL treatment had no effect on morphological apoptotic bodies and sub-G1 populations in normal human mesangial cells (MC) and normal human skin fibroblast (HSF) cells (Figure 7c). In addition, we did not distinguish survivin protein level in normal cells (Figure 7d). We also analyzed survivin and DR5 expression in renal clear cell carcinoma patient using TCGA databases through University of California Santa Cruz (UCSC) Xena Public Data Hub (xena.ucsc.edu) [28]. Survivin and DR5 is highly expressed in tumor (Appendix A). Moreover, high expression of survivin and DR5 is shorter the survival rate and showed poor prognosis (Appendix A). Therefore, these data indicate that IITZ-01 sensitizes to TRAIL-induced cell death in cancer cells.

## 3. Discussion

In this study, we demonstrated that lysosomotropic autophagy inhibitor, IITZ-01, enhanced TRAIL-mediated apoptosis in cancer cells, but not in normal cells. We found that stabilization of DR5 and degradation of survivin by IITZ-01 play a critical role in TRAIL-mediated apoptosis. Downregulation of Cbl E3 ligase is associated with DR5 upregulation in IITZ-01 treated cells. In addition, IITZ-01 decreased survivin stability via downregulation of USP9X expression. Therefore, we suggested that IITZ-01 could enhance TRAIL-induced apoptosis via modulation of DR5 and survivin expression (Figure 8).

IITZ-01, a novel potent lysosomotropic autophagy inhibitor, induced vacuolated appearance of cells due to deacidify lysosomes. IITZ-01 showed more than 10-fold potent autophagy inhibition compare with chloroquine [16]. As shown in Figure 1b, IITZ-01 also induced vacuolation in renal carcinoma Caki-1 cells. Previous study reported about the mechanism of IITZ-01 decreases IAP family protein involving cIAP1, XIAP and survivin in triple-negative breast cancer cells [16]. However, in our study, survivin was downregulated by IITZ-01, whereas expression of other IAP family protein (cIAP1, cIAP2 and XIAP) was not changed (Figure 2). This contradiction is believed to be due to the difference between the cell line used and the concentration of the IITZ-01.

DRs expression is critical for TRAIL-mediated apoptosis [29]. IITZ-01 increased DR5 protein level not DR4 (Figure 2). We reported the anti-cancer effect of lysosomotropic compounds (CQ, Monensin, and Nigericin) in renal carcinoma Caki-1 cells [23]. Other lysosomotropic compounds also increased DR5 expression. CQ (30 μM) induced both mRNA and protein of DR5 expression, while monensin, and nigericin only increased stability of DR5 protein. Interestingly, all drugs decreased Cbl expression. Although Cbl induces proteasome-dependent degradation via ubiquitination of target proteins, Cbl also could modulate lysosomal degradation of target proteins in a ubiquitin-dependent or -independent manner. For examples, Cbl mediates lysosomal degradation of K63-linked polyubiquitinating gp130 in interleukin 6-treated cells, and mono-ubiquitinating Notch1 in skeletal myoblast [30]. Cbl regulates lysosomal degradation of cystic fibrosis transmembrane conductance regulator in an ubiquitin-independent manner [31]. Furthermore, Cbl negatively regulates receptor-mediated signaling activation via lysosome-dependent degradation. After ligand engagement, internalized T cell receptor (TCR) is degraded by Cbl in lysosomes [32], and ligand binding EphB1 receptor is also Cbl-dependently degraded in lysosome [33]. In addition, Cbl negatively modulates activation of protein tyrosine kinase through the lysosomal pathway [26,27]. Although proteasome inhibitors upregulate DR5 expression [34], lysosome is also important for modulation of DR5 proteins expression [35]. In our study, since IITZ-01 induced DR5 expression in a Cbl-dependent manner (Figure 4c) and IITZ-01 is a lysosomotropic agent, it is a possibility that IITZ-01 inhibits lysosomal degradation of DR5.

Survivin contains only a single BIR domain and is one of the inhibitor of apoptosis (IAP) family proteins. Upregulation of survivin protein inhibit apoptosis both in a caspase-dependent and -independent way by interacting with XIAP and mitochondrial apoptosis-inducing factor (AIF), respectively [36,37]. Therefore, downregulation of survivin protein expression is targeted to induce cancer cell death. Here, we identified that IITZ-01 induced survivin expression via protein stabilization (Figure 5a,b). Degradation of survivin is mainly mediated by proteasome via ubiquitination [38]. E3 ligase XIAP regulates the degradation of survivin by ubiquitin-proteasome pathway [39]. However, IITZ-01 did not affect on expression levels of XIAP (Figure 2). Deubiquitinating enzymes (DUBs) can prevent protein degradation by removing ubiquitin from target substrate [40]. Previously, Chen et al. reported that USP9X is involved in deubiquitination and degradation of survivin by inhibition of long noncoding RNA LNC473 [41]. Furthermore, we reported that STAM-binding protein-like 1 (STAMBPL1) directly interacts and deubiquitinates survivin expression [42]. However, IITZ-01 only decreased USP9X expression and ectopic expression of USP9X prevented downregulation of survivin in IITZ-01 treated cells (Figure 6a,c). However, it need to further studies to identify the molecular mechanism of IITZ-01-induced downregulation of USP9X. Moreover, we found IITZ-01 downregulated USP9X through post-translation level (Figure 6b,c).

Taken all together, we suggest that ubiquitin-proteasome pathway-mediated DR5 upregulation and survivin downregulation have a critical role in lysosomotropic autophagy inhibitor (IITZ-01)-mediated sensitization of cancer cells to TRAIL-induced apoptosis.

## 4. Materials and Methods

### 4.1. Cell Cultures and Materials

Human renal carcinoma (Caki-1, ACHN and A498), human lung cancer (A549) and human breast cancer (MCF7) were procured from American Type Culture Collection (Manassas, VA, USA). Lonza (Basel, Switzerland) provided human mesangial cells, and Korea Cell Line Bank (Seoul, Korea) provided normal human skin fibroblasts cells. These cells were cultured in appropriate medium containing 10% fetal bovine serum, 1% penicillin/streptomycin and 100 μg/mL gentamicin at 37 °C in a humidified atmosphere with 5% CO_2_. IITZ-01 was purchased from Selleckchem (Houston, TX, USA). Human recombinant TRAIL and zVAD-fmk were provided by R&D system (Minneapolis, MN, USA). MG132 and lactacystin were supplied from Calbiochem (San Diego, CA, USA) and Enzo Life Sciences (Ann Arbor, MI, USA), respectively. Cycloheximide was provided from Sigma Chemical Co. (St. Louis, MO, USA). The primary antibodies were obtained as follows: anti-PARP, anti-cleaved caspase-3, anti-Bcl-xL, anti-DR5 from Cell Signaling Technology (Beverly, MA, USA). Anti-Bim, anti-Bax and anti-XIAP from BD Biosciences (San Jose, CA, USA). Anti-Mcl-1, anti-Bcl-2, anti-cIAP2, anti-Cbl from Santa Cruz Biotechnology (St. Louis, MO, USA). Anti-survivin from R&D system (Minneapolis, MN, USA). Anti-cIAP1 and anti-DR4 from Abcam (Cambridge, MA, USA), Anti-USP9X from Abnova (Taipei City, Taiwan). Anti-caspase-3 and anti-c-FLIP from Enzo Life Sciences (San Diego, CA, USA). Anti-actin from Sigma Chemical Co. (St. Louis, MO, USA). pCMV-Myc-Cbl plasmid was a gift from Dr. S. J. Kim (CHA University, Korea), and USP9X wild-type and USP9X/C1566S plasmids were a gift from Dr. D. S. Lim (KAIST, Korea).

### 4.2. Flow Cytometry Analysis

To analyze apoptosis, cells were harvested and fixed with 95% ethanol at least 1 h at 4 °C. Furthermore, cells were incubated in 1.12% sodium citrate buffer containing RNase at 37 °C for 30 min, added to 50 μg/mL propidium iodide, and analyzed using BD Accuri™ C6 flow cytometer (BD Biosciences, San Jose, CA, USA) [43].

### 4.3. Western Blotting

Cells have been lysed in RIPA lysis buffer (20 mM HEPES and 0.5% Triton X-100, pH 7.6) and supernatant fraction were collected. Proteins were separated by Sodium Dodecyl Sulfate-Polyacrylamide Gel Electrophoresis (SDS-PAGE) and transferred to the nitrocellulose membranes (GE Healthcare Life Science, Pittsburgh, PO, USA). Incubated with specific antibody and bands were detected using Immobilon Western Chemiluminescent horseradish peroxidase (HRP) Substrate (EMD Millipore, Darmstadt, Germany).

### 4.4. Annexin V and 7-AAD Staining

FITC-conjugated Annexin V and 7-aminoactinomycin D (7-AAD) (BD Pharmingen, San Jose, CA, USA) were used to estimate cell death mode. Cells were washed in cold PBS and resuspended in Annexin V-binding buffer. We added Annexin V-FITC and 7-AAD into the suspended cells, and then incubated for 15 min at room temperature in the dark. The cell death population was detected by flu BD Accuri™ C6 flow cytometer (BD Biosciences, San Jose, CA, USA).

### 4.5. DAPI, DNA Fragmentation Assay and Caspase Activity Assay

To investigate the nuclei condensation, cellular nuclei cells were stained with 300 nM 4′, 6′-diamidino-2-phenylindole solution (Roche, Mannheim, Germany), and we viewed fluorescence images using fluorescence microscopy (Carl Zeiss, Jena, Germany) [44]. To analyze DNA fragmentation, we used death detection ELISA plus kit (Boehringer Mannheim, Indianapolis, IN, USA) according to the manufacturer’s recommendations. To measure DEVDase activity, cells were treated with IITZ-01 and/or TRAIL, harvested, and incubated with reaction buffer containing acetyl-Asp-Glu-Val-Asp p-nitroanilide (Ac-DEVD-pNA) substrate, as previously mentioned [45].

### 4.6. Reverse Transcription-Polymerase Chain Reaction (RT-PCR) and Quantitative PCR (qPCR)

To isolate the total RNA, we used TriZol reagent (Life Technologies, Gaithersburg, MD, USA), and obtained cDNA using M-MLV reverse transcriptase (Gibco-BRL, Gaithersburg, MD, USA). For PCR, we used Blend Taq DNA polymerase (Toyobo, Osaka, Japan) with primers targeting DR5, Cbl, survivin and actin as mentioned in our previous studies [46,47]. For qPCR we utilize SYBR Fast qPCR Mix (Takara Bio Inc., Shiga, Japan) and reactions were performed on Thermal Cycler Dice^®^ Real Time System III (Takara Bio Inc., Shiga, Japan). The following primers were used for the amplification of DR5 and actin as described as our previous study [23]. We used actin as a reference Gene to calculate the threshold cycle number (Ct) of DR5 gene and reported the delta-delta Ct values of the genes.

### 4.7. Luciferase Activity Assay

The DR5 (-605) or DR5 (SacI) promoter-constructs transfected into the cells using Lipofectamine™ 2000 (Invitrogen, Carlsbad, CA, USA). Furthermore, cells were collected and harvested in lysis buffer (25 mM Tris-phosphate pH 7.8, 2 mM EDTA, 1% Triton X-100, and 10% glycerol). The supernatants were used to measure the luciferase activity according to the manufacturer’s instructions (Promega, Madison, WI, USA).

### 4.8. Transfection

For developing stable cell lines, Caki-1 cells were transfected with the control plasmid pcDNA 3.1(+) vector or pcDNA 3.1(+)/survivin-flag plasmids using Lipofectamine™2000 (Invitrogen, Carlsbad, CA, USA). After 24 h, cells were picked by 700 μg/mL G418 (Invitrogen, Carlsbad, CA, USA). For knockdown of gene, Caki-1 cells were transfected with the control siRNA (Bioneer, Daejeon, Korea), DR5 siRNA (Invitrogen, Carlsbad, CA, USA) or Cbl siRNA (Santa Cruz Biotechnology, St. Louis, MO, USA) using Lipofectamine^®^ RNAiMAX Reagent (Invitrogen, Carlsbad, CA, USA). Furthermore, protein expressions were checked by western blotting.

### 4.9. Immunoprecipitation Assay

Cells were collected, washed with PBS, lysed with RIPA lysis buffer containing 10 mM nethylmaleimide (NEM) (EMD Millipore, Darmstadt, Germany) and 1 mM PMSF, and then sonicated for protein extraction in ice. After sonication, cell lysates were centrifuged at 13,000× *g* for 15 min at 4 °C. The supernatants were incubated with 1 μg of anti-USP9X or anti-survivin antibody overnight at 4 °C, and then attached to 20 μL of Protein G agarose bead using the rotator at 4 °C for 2 h. Cell lysates were washed with RIPA lysis buffer containing 10 mM NEM and 1 mM PMSF, and boiled in 2× sample buffer for 10 min. Protein–protein interactions were checked by Western blotting.

### 4.10. Ubiquitination Assay

The assay was performed as described in our previous study [45]. Cells were co-transfected with HA-tagged ubiquitin (HA-Ub), USP9X/wild type and UPS9X/C1566S plasmid, and treated with MG132 for 12 h. Immunoprecipitation was performed using the anti-survivin, and ubiquitination of endogenous survivin was checked using HRP-conjugated anti-Ub under denaturing conditions.

### 4.11. Statistical Analysis

The data were analyzed using a one-way ANOVA and post-hoc comparisons (Student-Newman-Keuls) using the Statistical Package for Social Sciences 22.0 software (SPSS Inc.; Chicago, IL, USA).

## 5. Conclusions

Our study suggests that IITZ-01, a lysosomotropic agent, sensitizes TRAIL-induced apoptosis via DR5 upregulation and survivin downregulation in cancer cells, but not normal cells. In mechanisms, IITZ-01 induces DR5 stabilization by downregulation of Cbl and USP9X-dependent survivin ubiquitination and degradation.

## Figures and Tables

**Figure 1 cancers-12-02363-f001:**
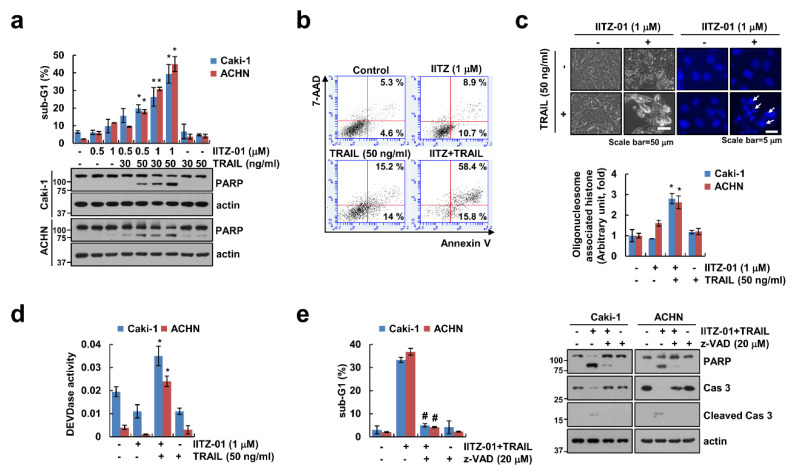
IITZ-01 sensitizes renal carcinoma cells to TNF-related apoptosis-inducing ligand (TRAIL-induced apoptosis. (**a**) Caki-1 and ACHN cells were incubated with indicated concentrations with IITZ-01 and/or TRAIL for 18 h. The apoptotic population (sub-G1) and protein levels were analyzed by flow cytometry and Western blotting, respectively; (**b**,**c**) Caki-1 cells were incubated with 1 μM IITZ-01 and/or 50 ng/mL TRAIL for 18 h. Cell death was detected using Annexin V/7-AAD staining (**b**). The cell morphology was examined using interference light microscopy. Nuclei condensation and DNA fragmentation were detected using 4′,6′-diamidino-2-phenylindole (DAPI) staining and DNA fragmentation detection kit, respectively (**c**); (**d**) Caki-1 and ACHN cells were incubated with 1 μM IITZ-01 and/or 50 ng/mL TRAIL for 18 h. DEVDase (caspase-3) activity was examined using kit as a described in a Material and Methods; (**e**) Caki-1 and ACHN cells were incubated with 1 μM IITZ-01 plus 50 ng/mL TRAIL in the presence or absence of 20 μM z-VAD-fmk (z-VAD). The values in graph (**a**,**c**–**e**) represent the mean ± SD of three independent experiments. * *p* < 0.05 compared to the control. # p < 0.05 compared to IITZ-01 plus TRAIL.

**Figure 2 cancers-12-02363-f002:**
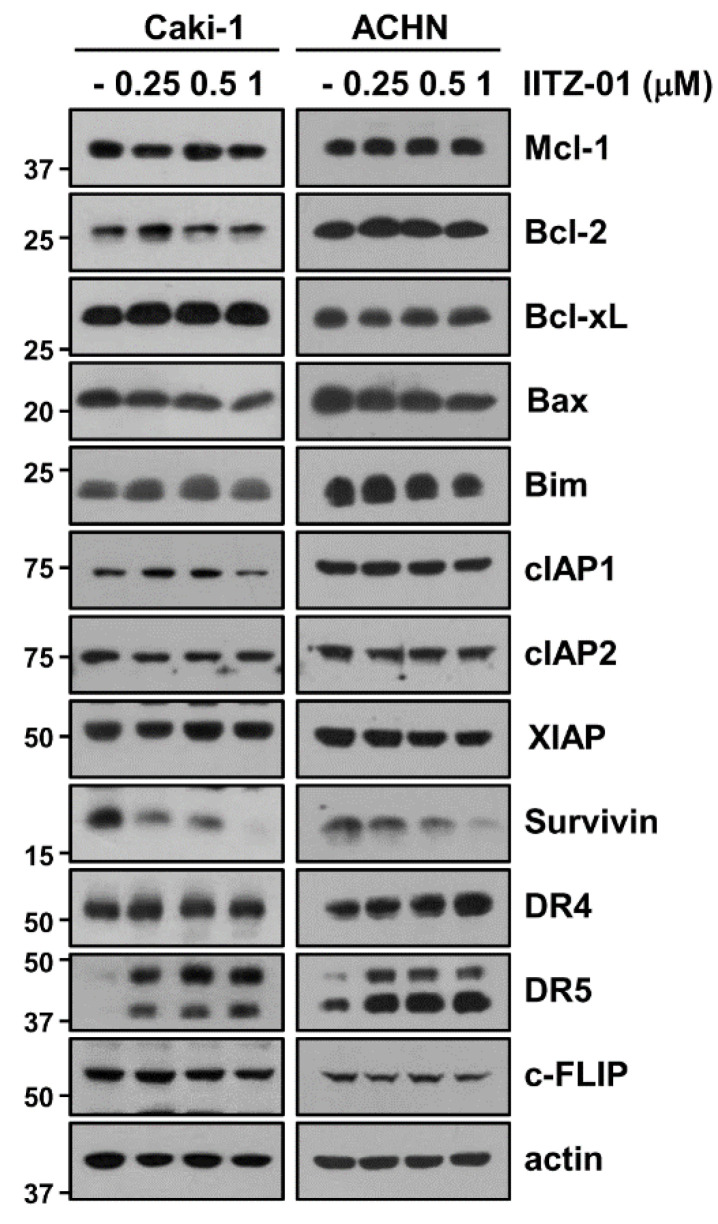
IITZ-01 induces upregulation of DR5 and downregulation of survivin. Caki-1 and ACHN cells were incubated with 0.25–1 μM IITZ-01 for 18 h. The protein expression levels of apoptosis related proteins and actin were determined by Western blotting.

**Figure 3 cancers-12-02363-f003:**
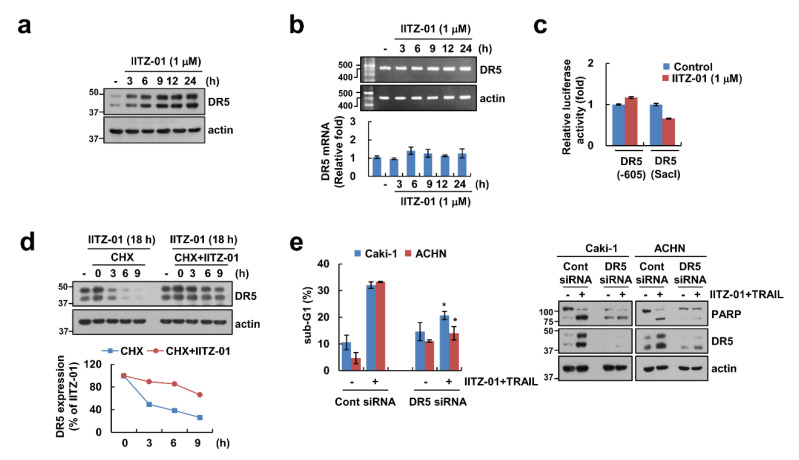
DR5 stabilization contributes to the sensitization effect of IITZ-01 on TRAIL-induced apoptosis in Caki-1 cells. (**a**,**b**) Caki-1 cells were incubated with 1 μM IITZ-01 for indicated time periods. The protein and mRNA levels were determined by Western blotting (**a**) and RT-PCR (upper panel)/qPCR (low pane), respectively (**b**); **(c)** Caki-1 cells were transiently transfected with DR5(-605) or DR5(SacI) promoter and incubated with 1 μM IITZ-01 for 18 h. The cells were lysed and the luciferase activity was measured as a described in a Material and Methods; (**d**) Caki-1 cells were incubated with 1 μM IITZ-01 for 18 h, washed with PBS, and then treated with 20 μg/mL cycloheximide (CHX) and/or 1 μM IITZ-01 for the indicated time periods. The band intensity of the DR5 protein was measured using ImageJ; (**e**) Caki-1 and ACHN cells were transiently transfected with control (Cont) siRNA or DR5 siRNA, and then incubated with 1 μM IITZ-01 and 50 ng/mL TRAIL for 18 h. The apoptotic population (sub-G1) and protein levels were analyzed by flow cytometry (**e**) and Western blot (**d**,**e**). The values in graph (**b**,**c**,**e**) represent the mean ± SD of three independent experiments. * *p* < 0.05 compared to the IITZ-01 plus TRAIL in control siRNA.

**Figure 4 cancers-12-02363-f004:**
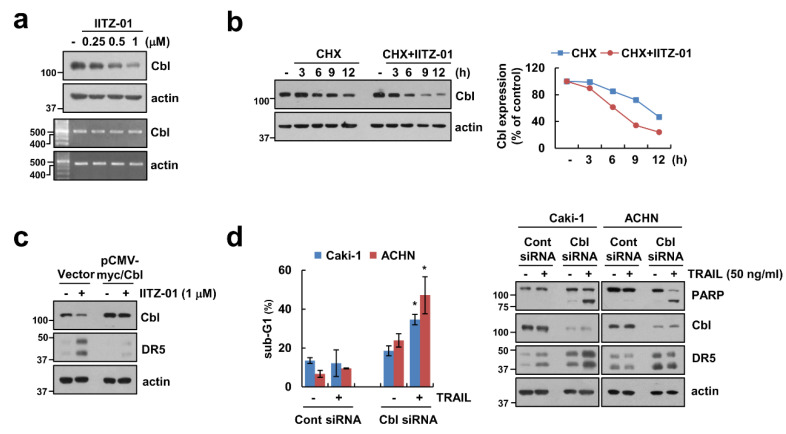
Downregulation of Cbl is involved in IITZ-01-mediated DR5 upregulation and TRAIL sensitization. (**a**) Caki-1 cells were incubated with 0.25–1 μM IITZ-01 for 18 h. The protein and mRNA levels were determined by Western blotting and RT-PCR, respectively; (**b**) Caki-1 cells were incubated with 20 μg/mL CHX and/or 1 μM IITZ-01 for the indicated time periods. The band intensity of the Cbl protein was measured using Image J; (**c**) Caki-1 cells were transfected with pCMV-myc (Vector) or pCMV-myc/Cbl and treated with 1 μM IITZ-01 for 18 h; (**d**) Caki-1 and ACHN cells were transiently transfected with Cont siRNA or Cbl siRNA, and then incubated with 50 ng/mL TRAIL for 18 h. The apoptotic population (sub-G1) and protein levels were analyzed by flow cytometry (**d**) and Western blotting (**b**–**d**)**,** respectively. The values in graph (**d**) represent the mean ± SD of three independent experiments. * *p* < 0.05 compared to the TRAIL in control siRNA.

**Figure 5 cancers-12-02363-f005:**
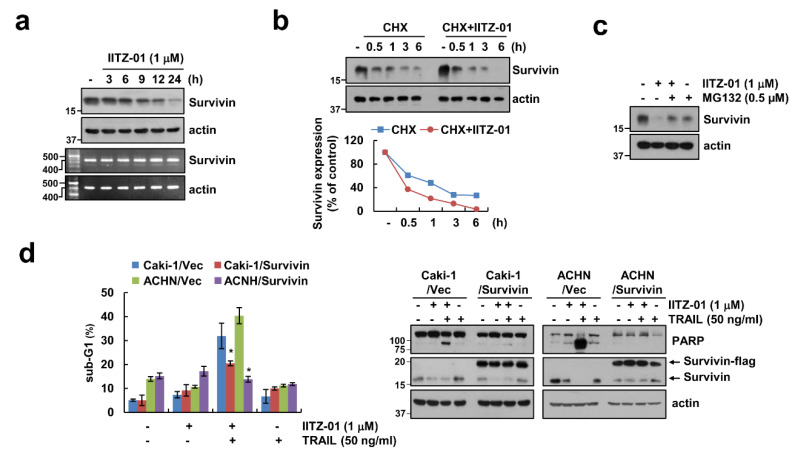
Survivin downregulation contributes to the sensitization effect of IITZ-01 on TRAIL-induced apoptosis in Caki-1 cells. (**a**) Caki-1 cells were incubated with 1 μM IITZ-01 for indicated time periods. The protein and mRNA levels were determined by Western blotting or RT-PCR, respectively; (**b**) Caki-1 cells were incubated with 20 μg/mL CHX and/or 1 μM IITZ-01 for the indicated time periods. The band intensity of the survivin protein was measured using Image J; (**c**) Caki-1 cells were pretreated with 0.5 μM MG132, and then incubated with 1 μM IITZ-01 for 18 h; (**d**) Vector-transfected cells (Caki-1/Vec and ACHN/Vec) and survivin-overexpressing cells (Caki-1/Survivin and ACHN/Survivin) were incubated with 1 μM IITZ-01 and/or 50 ng/mL TRAIL for 18 h. The values in graph (**b**,**d**) represent the mean ± SD of three independent experiments. * *p* < 0.05 compared to the IITZ-01 plus TRAIL in Vector cells.

**Figure 6 cancers-12-02363-f006:**
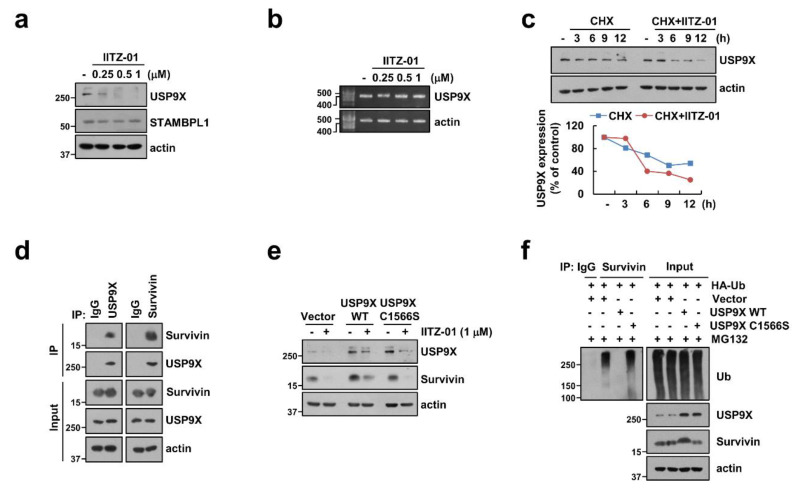
Degradation of survivin is critical to induction of IITZ-01 plus TRAIL-induced apoptosis through downregulation of USP9X. (**a**,**b**) Caki-1 cells were incubated with 0.25–1 μM IITZ-01 for 18 h. USP9X mRNA levels were determined by RT-PCR; (**c**) Caki-1 cells were incubated with 20 μg/mL CHX and/or 1 μM IITZ-01 for the indicated time periods. The band intensity of the USP9X protein was measured using Image J; (**d**) The interaction with survivin and USP9X was indicated by immunoprecipitation (IP) assay; (**e**) Caki-1 cells were transiently transfected with pDESTS1 (Vector), pDESTS1-USP9X (USP9X WT) or pDESTS1-USP9X/C1566S (USP9X C1566S), and then incubated with 1 μM IITZ-01 for 18 h; (**f**) To analyze the ubiquitination of endogenous survivin, Caki-1 cells were transfected with HA-ubiquitin (HA-Ub), USP9X WT and USP9X C1566S and treated with 0.5 μM MG132. Cells were lysed in 1% sodium dodecyl sulfate (SDS) buffer to disrupt interacting proteins and cell lysates were then diluted to 0.1% SDS, followed by the immunoprecipitation using an anti-survivin. Survivin ubiquitination was detected by Western blotting using an horseradish peroxidase (HRP)-conjugated anti-Ub antibody. The protein levels were analyzed by Western blotting (**a**,**c**–**f**).

**Figure 7 cancers-12-02363-f007:**
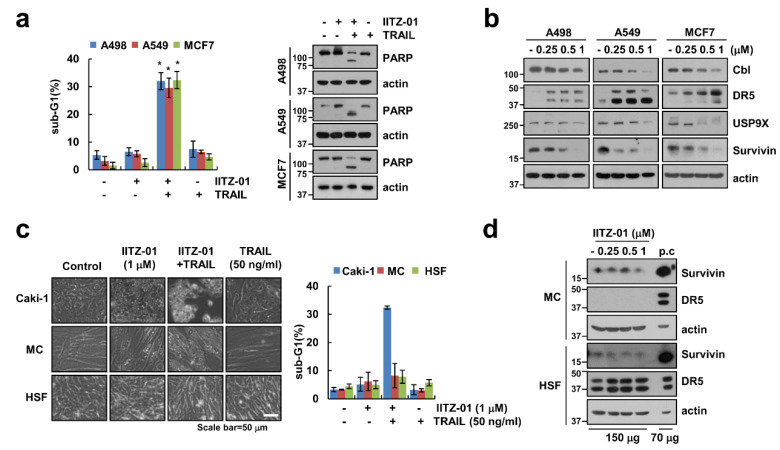
IITZ-01 plus TRAIL induces apoptosis on other carcinoma cells, but not normal cells. (**a**) Renal carcinoma (A498), lung carcinoma (A549) and breast carcinoma (MCF7) were incubated with 1 μM IITZ-01 and/or 50 ng/mL TRAIL for 18 h; (**b**) A498, A549 and MCF7 cells were incubated with 0.25–1 μM IITZ-01 for 18 h; (**c**) Caki-1, human renal normal mesangial cells (MC) and human skin fibroblast (HSF) cells were incubated with 1 μM IITZ-01 and/or 50 ng/mL TRAIL for 18 h. The cell morphology was examined using interference light microscopy; (**d**) MC and HSF cells were incubated with 0.25–1 μM IITZ-01 for 18 h. (positive control (p.c); Caki-1 cell lysate). The apoptotic population (sub-G1) and protein levels were analyzed by flow cytometry (**a**,**c**) and Western blotting (**a**,**b**,**d**), respectively. The values in graph (**a**,**c**) represent the mean ± SD of three independent experiments. * *p* < 0.05 compared to the control.

**Figure 8 cancers-12-02363-f008:**
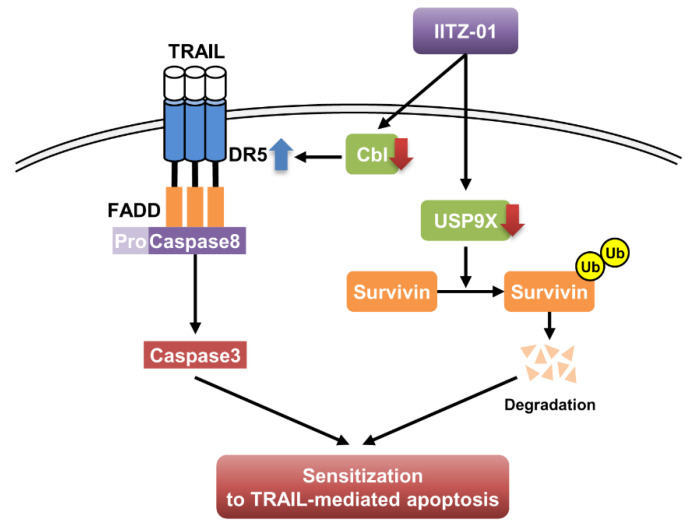
The diagram indicating the mechanism of IITZ-01-induced TRAIL sensitization.

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
