# Peer review of "Upregulation of DR5 and Downregulation of Survivin by IITZ-01, Lysosomotropic Autophagy Inhibitor, Potentiates TRAIL-Mediated Apoptosis in Renal Cancer Cells via Ubiquitin-Proteasome Pathway"

_cancers, 2020, doi:10.3390/cancers12092363_

Round 1

Reviewer 1 Report

To the Authors

In this original article, Shahriyar et al. are discussing a mechanistic regulation of TRAIL-mediated apoptosis on renal cancel cell lines after treatment with the lysosomotropic autophagy inhibitor IITZ-01.

Cancer cells can exhibit different levels of resistance to TRAIL-mediated apoptosis. Anyway, sensitizing agents to this modality of activation of regulated cell death have been described in the literature. The Authors showed that IITZ-01 (a compound known to accumulate autophagosomes and reduce mitochondrial membrane permeabilization) can sensitize kidney cancel cell lines to TRAIL-mediated apoptosis through diverse mechanisms affecting DR5 and Survivin expression levels. Moreover, the effect of IITZ-01 on cancer cell lines does not seem to be active on normal cells, therefore suggesting this treatment as selective towards malignant cells only.

I personally think this work is quite linear but I have a Major concern and a few minor concerns on the experiments shown.

Major concern:

  • Once the Authors demonstrated that the cell death modality induced by TRAIL after IITZ-01 treatment is caspases-dependent (Figure 1), they focus on many proteins involved in the apoptosis process, recognising an increase of DR5 and a decrease of Survivin expression levels (Figure 2). These experiments have been carried out on Caki and ACHN cell lines but the experiments showing a mechanistic regulation of the two proteins are carried out just on the Caki cell line. This might undervalue the results obtained, since they might be dependent on the genetic background of the cells. I would therefore strongly recommend repeating DR5 and Cbl silencing also on ACHN (and/or on A498) as well as an overexpression of Survivin (basically referring to experiments shown in panels 3E, 4D and 5D).

Minor concerns:

  • Figure 1 – I personally think an Annexin V/Propidium Iodide analysis should be mandatory to talk about apoptosis.
  • In the experiments the Authors are using Caki cell lines. Which of the two existing cell lines? Caki-1 or Caki-2?
  • I do not totally agree with the statement that the IITZ-01 does sensitize cancer cells but not normal cells to TRAIL-mediated apoptosis. Panel 7D shows the effect of IITZ-01 on MC and HSF normal cell lines. Anyway, Survivin protein level is not detectable in MC and HSF cells and therefore we cannot conclude that IITZ-01 does not have a potential effect similar to the one exerted on cancer cells. How would the Authors explain this?
  • To further increase the impact of the results obtained, I would suggest performing a bioinformatic analysis on the expression levels of DR5, Cbl, Survivin an USP9X in renal cancer patients form one of the major existing databases (TCGA as an example).

Reviewer 2 Report

In this study the authors provide with interesting data regarding the mechanism by which the autophagy inhibitor IITZ-01 enhances TRAIL mediated apoptosis in renal cancer cells. The methods and results obtained are well described and support the conclusions, providing with novel mechanistic data. It is also of merit that, in addition to the kidney cancer cell line Caki, additional cell models are included in the last part of the results section. Some comments are provided below.

  1. While the downregulation of Cbl by IITZ-01 is demonstrated to occur at protein level with CHX treatment, little is commented on USP9X downregulation by IITZ-01. Little is known about the regulation of the expression of this protein, and in cancer cells both increased and decreased expression has been detected compared with normal tissues (e.g. 3978/j.issn.2072-1439.2015.04.28, 10.3892/or.2019.7131). In addition, the downregulation of USP9X has been associated with both poor and good prognosis (e.g. references above, 10.1007/s11845-020-02199-2, 10.3892/ol.2018.8452). Without the need to perform an exhaustive work, it would be interesting to provide with some additional details regarding the downregulation of USP9X in this cell model.

  1. Comparison of TRAIL sensitivity between cancer and non-cancer cell lines are performed in the study. Given that sensitivity to TRAIL&IITZ-01 relied on survivin downregulation, an explanation of the few subG1 cell levels in normal cells in this condition is lacking. Taking into account the undetectable levels of survivin in these cells, is DR5 induced by IITZ-01 treatment? Do these non-cancer cells display lower levels of DR5? MeasuringDR5 levels in these cells could reinforce DR5-survivin role in TRAIL sensitization.

Round 2

Reviewer 1 Report

I sincerely have to apologize.
Anyway, my review would have been positive since the authors have answered to all my concerns adding the requested and pertinent additional experiments.
I am very sorry about my delay, due to vacation, but I would anyway have expressed a positive comment on the paper (which I have carefully read in this revised version).